# The Relation between Physiological Parameters and Colour Modifications in Text Background and Overlay during Reading in Children with and without Dyslexia

**DOI:** 10.3390/brainsci11050539

**Published:** 2021-04-25

**Authors:** Tamara Jakovljević, Milica M. Janković, Andrej M. Savić, Ivan Soldatović, Gordana Čolić, Tadeja Jere Jakulin, Gregor Papa, Vanja Ković

**Affiliations:** 1Jožef Stefan International Postgraduate School, 1000 Ljubljana, Slovenia; 2School of Electrical Engineering, University of Belgrade, 11000 Belgrade, Serbia; piperski@etf.rs (M.M.J.); andrej_savic@etf.rs (A.M.S.); 3Institute of Medical Statistics and Informatics, Faculty of Medicine, University of Belgrade,11000 Belgrade, Serbia; ivan.soldatovic@med.bg.ac.rs; 4The College of Social Work, 11000 Belgrade, Serbia; gordana.colic22@gmail.com; 5Faculty of Tourism Studies, University of Primorska, 6320 Portorož, Slovenia; tadeja.jerejakulin@upr.si; 6Computer Systems Department, Jožef Stefan Institute, 1000 Ljubljana, Slovenia; gregor.papa@ijs.si; 7Laboratory for Neurocognition and Applied Cognition, Faculty of Philosophy, University of Belgrade, 11000 Belgrade, Serbia; vanja.kovic@f.bg.ac.rs

**Keywords:** dyslexia, reading, children, background colour, overlay colour, text colour, sensors, physiological parameters, EEG, ECG, EDA, eye tracking

## Abstract

Reading is one of the essential processes during the maturation of an individual. It is estimated that 5–10% of school-age children are affected by dyslexia, the reading disorder characterised by difficulties in the accuracy or fluency of word recognition. There are many studies which have reported that coloured overlays and background could improve the reading process, especially in children with reading disorders. As dyslexia has neurobiological origins, the aim of the present research was to understand the relationship between physiological parameters and colour modifications in the text and background during reading in children with and without dyslexia. We have measured differences in electroencephalography (EEG), heart rate variability (HRV), electrodermal activities (EDA) and eye movements of the 36 school-age (from 8 to 12 years old) children (18 with dyslexia and 18 of control group) during the reading task in 13 combinations of background and overlay colours. Our findings showed that the dyslexic children have longer reading duration, fixation count, fixation duration average, fixation duration total, and longer saccade count, saccade duration total, and saccade duration average while reading on white and coloured background/overlay. It was found that the turquoise background, turquoise overlay, and yellow background colours are beneficial for dyslexic readers, as they achieved the shortest time duration of the reading tasks when these colours were used. Additionally, dyslexic children have higher values of beta (15–40 Hz) and the broadband EEG (0.5–40 Hz) power while reading in one particular colour (purple), as well as increasing theta range power while reading with the purple overlay. We have observed no significant differences between HRV parameters on white colour, except for single colours (purple, turquoise overlay, and yellow overlay) where the control group showed higher values for mean HR, while dyslexic children scored higher with mean RR. Regarding EDA measure, we found systematically lower values in children with dyslexia in comparison to the control group. Based on the present results, we can conclude that both pastel and intense background/overlays are beneficial for reading of both groups and all sensor modalities could be used to better understand the neurophysiological origins in dyslexic children.

## 1. Introduction

It is estimated that more than 10 percent of the world population is affected by dyslexia. Dyslexia is a learning disability of neurobiological origin whose main characteristics are difficulties with accurate or fluent word recognition. It is also characterized by poor spelling and decoding abilities which results in problems in reading comprehension and reduced reading experience [1]. Developmental dyslexia is manifested by individuals who require special motivation and intellectual effort to achieve fluent reading and it is usually defined as an unexpected difficulty in reading [2,3]. The World Federation of Neurology recognizes children with dyslexia as those who, despite the common school curriculum, cannot attain reading, writing, and spelling skills in proportion to their intellectual abilities [4,5]. Although definitions and understandings of dyslexia vary, there is a general opinion that children who fail to accurately read, need careful support and monitoring at an early school age. At that age, the most effective approach for children with reading difficulties are early identification and professional support which assists their needs [6,7,8]. For example, in Sweden, children are being diagnosed with dyslexia on average only at the age of thirteen [4]. At the early-school age, it is not so difficult to mark the grade level of reading, but it is difficult to understand other problems which usually impede overall school achievements and cause emotional distress and lack of motivation [9,10]. Children who have this type of reading difficulty do not have difficulties in their general performance in other segments of the curriculum. Some of them may also have emotional difficulties [11,12,13]. Through early intervention, the risk of those difficulties may be reduced before they emerge in the fourth grade. In the research dedicated to the early indicators of dyslexia, children were monitored from preschool age to the end of the fourth grade of primary school [14]. It is found that indicators of dyslexia can be isolated at preschool age. There is evidence that dyslexia can be prevented and predicted in early school-age children [15,16,17]. Today, scientists are better able to understand the child’s nervous system and how it operates in the case of dyslexia using functional neuroimaging [18].

There is good evidence that dyslexia has a neurological basis [19,20]. Namely, it may be reflected in the psycho-physiological states of the body during the reading task.

Given that colours can affect the state of our body and emotions [21,22], this study investigates the relationship between physiological parameters and colour modifications in the text and background during reading in children with and without dyslexia. It is evident that colours have been used in modulating reading performance in children with dyslexia in order to improve their skills and for other purposes such as increasing reading fluency and speed [23,24,25,26,27]. The so-called visual stress syndrome is observed in dyslexic individuals very often [28,29], but the role of colour involvement during the reading performance remains controversial [30,31,32,33,34]. Recently, this concept has led to the broadened use of coloured overlays to mitigate reading disorders and improve reading. It is reported that symptoms of visual stress are caused by sensitivity to certain light frequencies [35]. In practice, applications of coloured filters have led to the use of lenses and overlays in reading. Coloured overlays filter the light using transparent plastic reading sheets tinted with colour over the text [36]. Many dyslexic subjects reported that coloured overlays can help them with widespread difficulties arising in the reading process [37,38,39,40,41]. Moreover, it has been reported, without taking into consideration children with dyslexia, that colour overlays could improve the reading process in school-age children [42]. Furthermore, other scientists [39,43] have shown that problems caused by dyslexia could be relieved by visual changes in the presentation of the reading text. Based on these studies, others have focused on designs of computer screen texts, where parameters such as background, text colour, or font size have been adjusted in order to help those with dyslexia [44,45]. These conditions inspired Pinna and Deiana [46,47] to study how reading time and comprehension could be influenced by colours.

Other studies taking into account the effects of colour on reading in dyslexic children showed possibilities of reduction of symptoms in Meares Irlen Syndrome by application of colour filters which eased the visual discomfort in these children [48]. Furthermore, coloured filters were then considered as an effective intervention for delayed readers who had experienced visual stress. It was reported [49] that the reading speed of patients with Meares-Irlen syndrome improved by more than 20% when they wore the selected colour-tinted lenses. By contrasting no filter, yellow, and green filter, researchers [50,51] found that children with dyslexia were fastest and had the shortest fixation time when reading in the yellow and green-filter condition.

Recently, Stein [52] came up with the argument that dyslexia is “characterized by poor temporal processing, hence impaired visual and auditory sequencing, that is caused by impaired development of transient/magnocellular (M-) systems throughout the brain,” and thus it is necessary to collect evidence not only from psychophysical tests, but from electrophysiological, eye movement, attentional, brain imaging, interventional, and genetic findings as well. Aside from the use of colour overlays, it was also found that the use of different background colours [44] increased reading performance of subjects with dyslexia and has been recommended by the British Dyslexia Association [53].

Reading involves sensory integration, attention, and memory processes which are reflected in the psycho-physiological states of the individual and measurable by different biosignals. Our previous work introduced a sensorhub for measurement of four biosignal modalities and describes the contributions of each modality to understanding of different aspects of neural, physiological, and behavioral processes in children during reading [21].

Based on the previous studies [11,54] and results, the main purpose of the present study is to extend previously established experimental protocol to children with dyslexia and to better understand the neurological basis of this disorder and its relation to colours.

The current study presents the first research that explores the impact of 13 combinations of background and overlay colours on the reading performance of children with normal reading skills and children diagnosed with dyslexia, measured by multimodal sensor hub (electroencephalography, EEG; electrocardiography, ECG; electrodermal activity, EDA; and eye-tracking).

We expected that children with good reading skills would be less affected by the change of colour. We also expected more variation in reading performance with the introduction of coloured elements in dyslexic children. However, in both groups, we expected that cognitive and emotional arousal would vary with the change of colour and that could only be measured by employing fine-grained measurement tools relying on automatic measures of cognitive and emotional engagement in the task. We expected that the pastel colours would facilitate the reading task, unlike intense colours, which we expected would be more challenging both to dyslexic and non-dyslexic children, based on previous reports [44,50,51]. These parameters make it possible to conclude which colours would result with the better focus on the text, which otherwise could not be assessed with rough measures such as reading duration.

This study is a step towards defining the set of measurements which may enable the quick automatic selection of colour setup that would facilitate the reading task in a specific reader.

## 2. Materials and Methods

### 2.1. Subjects

Thirty-six participants took part in this study (18 with dyslexia and 18 without dyslexia, matched according to gender and school grade). Children in the control group were randomly chosen from the second to sixth grade (8–12 years old) of three elementary schools in Belgrade. Children with dyslexia were selected from several elementary schools in Belgrade in coordination with a certified speech therapist. The dyslexic children in this study have already undergone adequate speech treatment for dyslexia. All of them satisfied the standard criteria for dyslexia diagnosed by the speech therapist based on three different tasks: speed of reading, accuracy (phonological decoding), and understanding measured on the standardized text adapted for their age in Serbian. In addition, the IQ of children was tested on the Raven’s Progressive Matrices and only those who scored higher than 90 IQ were selected for the study.

Every child from the group of dyslexic children was diagnosed with dyslexia by the specialist in the field and was included in the study based on that criterion. Secondly, they needed to have a normal or corrected to normal vision, meaning that they either had no eye-sight problem, or if they did, they had adequate glasses so that they could read the text with no difficulty. In fact, only one child with corrected vision took part in the study. All the other children had no eye-sight difficulties. For the control group of children, the inclusion criteria were that they have normal or corrected to normal vision and no learning and reading disabilities or attention disorders (as assessed by a certified speech therapist or their teachers). The exclusion criterion for both groups was presence of large artifacts in the acquired signals. In our sample, no such cases were observed. Also, no participants were excluded from the dataset for the statistical analysis, according to these criteria. The experiment was conducted in a classroom at the Faculty of Philosophy, University of Belgrade, where children participated individually under the same experimental conditions. Every child had received instructions from the researcher before the experiment (to read in silence, how to position their head at the chin rest, how to look at the external monitor etc.). After the reading test, the participants received a present (sticker and chocolate) and diploma. The collected data were fully anonymized. Only team members had access to the grade and gender of the children participating in the study. The research team collected informed consent from the parents for the children’s participation through speech therapists or school directors and teachers.

The experimental procedure was approved by the ethical committee of the Psychology Department of the University of Niš (a branch of the Serbian Psychology Association) No 9/2019.

### 2.2. Experiment Setup

During the experiment, participants were seated in front of the computer screen with a keyboard at the table, placing their head on the chinrest, making sure they were holding the same distance from the monitor. After the participants received instructions from the researcher, they read the story presented on the computer screen in silence. The reading text was selected from the textbook for the third grade, so that it did not contain any of the words that children would not be familiar with. Also, this text was selected because the paragraphs were comparable based on the length and complexity (per slide). Stimuli presentation was launched by pressing the space button (self-paced reading). The stimuli presentation started with a paragraph with black text on a white background (as a referent slide), as school-age children are used to in everyday life. After the referent slide, which would always appear at the beginning of the presentation, the following slides were presented in a pseudo-randomized order of background/overlay colours. Background colours were always presented with black text and overlays according to colour calculation in the section Experiment Design of the previous study [21] (marked by O in further text, e.g., blue O stands for blue overlay). The story was divided into 13 paragraphs/slides, so the text on each slide was kept in the original order but in different colours (in order to avoid the effect of factors such as semantic or affective content, vocabulary, text complicity, or syntax). Thus, we did pseudo-randomisation of the colours so that the same overlay and background/text colour would not appear one after the other, but every paragraph would be seen by different subjects on the different background colour. This way, we made sure that the particular differences, such as word frequency, length of the word, number of words, etc., would be averaged out.

### 2.3. Experiment Design

Experimental design was exactly the same as in the [21] except for the difference laid out in the description of Figure 1 and Data Processing part.

Figure 1 shows the sensor hub consisting of a portable multimodal ECG/EEG/EDA and eye-tracking system for acquisition of physiological data during the reading performance. A mobile 24-channel EEG amplifier (SMARTING, mBrainTrain, Belgrade, Serbia) was used for EEG and ECG signals recording. EDA recording was performed using a research prototype device [55]. SMI RED-m 120-Hz portable remote eye tracker (iMotions, Copenhagen, Denmark) was used for eye-tracking (9-point calibration was used). The SMI software was used for stimuli presentation (Experiment Centre 3.7), data collection (iViewRED-m), and data analysis and visualisation (BeGaze 3.7). Real-time monitoring and storage were at one laptop instead of two, which was the case in the previous experiment design [22] because the system was synchronized by keyboard stream instead of the photosensitive sensor. The laptop was connected with an external keyboard and external monitor positioned in front of every child. The sampling rate was set to 250 Hz for EEG/ECG data and 40 Hz for EDA data. The application for data acquisition used Lab Streaming Layer (LSL) for the synchronization between the EDA, EEG/ECG data, and keyboard stream, and it stored all data in XDF file format.

#### 2.3.1. Data Processing

(1) Extracted eye-tracking parameters from SMI system were: fixation count, fixation frequency (count/second), fixation duration total (ms), fixation duration average (ms), saccade count, saccade frequency (count/second), saccade duration total (ms), and saccade duration average (ms).

(2) Offline EEG processing was applied on the dataset of all subjects. EEG signals of all subjects and channels were band-pass filtered (4th order Butterworth filter) to extract the EEG activity in the following 5 frequency ranges: (a) delta (0.5–4 Hz), (b) theta (4–7 Hz), (c) alpha (7–13 Hz), (d) beta (15–40 Hz), and (e) broadband EEG activity (0.5–40 Hz).

Filtered signals of all channels were squared and segmented according to the keyboard event markers. Each epoch was associated with the reading of a single slide. For each subject, electrode site, frequency band, and epoch, the median value of filtered and squared EEG was calculated in order to obtain single band-power values for each epoch. Calculating the median band-power over epoch duration is used to remove impulse-noise due to movements, blinks, or other artefacts. Additional visual inspection of power epochs was conducted in order to ensure that the obtained median values represented the valid quantification of the band-power activity. Median power values of each slide/band were normalized by calculating the relative change using the following equation:(1)Pc=(medP−medPt)/medPt
where *Pc* is the measure of power change for each band/slide, *medP* is the median value of power for band/slide, and *medPt* the median value of power in frequency band for the whole recording. Measurement unit of *medPt* and *medP* is µV^2^ while *Pc* is unitless.

(3) Offline heart-signal processing for extraction of heart-rate variability (HRV) parameters in the time domain: (a) Mean value of beat-to-beat intervals (BBIs), mean RR (ms); (b) standard deviation of normal BBIs, SDNN(ms); (c) mean value of heart rate, mean HR (beats/min); (d) standard deviation of heart rate, STD HR (beats/min); (e) coefficient of variance of normal BBIs, CVRR=SDNN/mean RR (n.u.); and (f) Root mean square of differences of successive BBIs, RMSSD(ms).

(4) The average value of electrodermal activity was calculated for each slide.

#### 2.3.2. Statistical Analysis

Results are presented as count (percent), means ± standard deviation, or median (25–75th percentile), depending on data type and distribution. Parametric tests were used for the analysis of continuous variables with normal distribution, while non-parametric tests were used in analysis with non-normally distributed data. Normality of distribution was explored using descriptive statistics (mean, standard deviation, median, interquartile range, skewness, and kurtosis), tests of normality (Kolmogorov–Smirnov and Shapiro–Wilks), and graphical methods (QQ plot, histogram, boxplot).

When we compared groups by parameters per each colour, the sample consisted of 18 participants × 2 groups (dyslexic vs. non-dyslexic); the majority of samples skewed distribution, and, for that reason, we used the Mann–Whitney U test instead of independent samples *t* test (Table 1 and Table 2).

Data of measurements when all colours were gathered and the sample consisted of 18 participants × 13 colours × 2 groups (dyslexic vs. non-dyslexic) had a normal distribution with a sufficient number of samples. In this situation, we used an independent samples t-test (Table 3).

Additionally, in each group, separately, we compared measurement of each parameter on every single colour to white (difference between 12 colours and white colour) using paired-samples t-test (for differences with normal distribution) and Wilcoxon Signed Ranks test (for differences with non-normal distribution) (Appendix A).

This is an exploratory study that included several different sensor measurements (eye-tracking, EEG, EDA, HRV) for a better understanding of the neurophysiological origin of dyslexia and its relation to colours. No sample size prior to the study was calculated. Due to the pilot nature of the study, no *p*-value adjustment for multiple outcomes was applied.

All *p*-values which were less than 0.05 were considered significant. The data were analysed within the SPSS 20.0 software (IBM Corp. Released 2011. IBM SPSS Statistics for Windows, Version 20.0. Armonk, NY, USA: IBM Corp).

## 3. Results

### 3.1. White (Default) Background—Reading Performance Results

Reading performance (non-dyslexic vs. dyslexic) and physiological parameters measured by the sensor hub on a white background colour are presented in Table 1. For all of the eye-tracking measures, we have found a significant difference between groups, except for the fixation duration average. Dyslexic children showed significantly higher scores in comparison to non-dyslexic children in all eye-tracking measures, except for fixation frequency and saccade frequency. A significant difference was also observed with the EDA parameter, where non-dyslexic children had higher values in comparison to dyslexic children. In all HRV parameters, we observed no significant difference between dyslexic and non-dyslexic children. In mean HR and STD HR, non-dyslexic children achieved higher values, while dyslexic children showed higher scores in all other HRV parameters. Regarding EEG power bands, we found no significant differences between groups. In all EEG measures, dyslexic children showed higher values. Due to the distortion of the normal distribution of the data, the parameters are presented as medians and interquartile range and compared using a non-parametric alternative to the t test, the Mann–Whitney U test.

### 3.2. Modifications in Background and Overlay Colours—Reading Performance Results

In Table 2, we show median values for the overall results for each parameter per single colour. Dyslexic and non-dyslexic children were compared on each of the parameters, namely, reading duration, eye-tracking measures, EDA, HRV, and EEG parameters. The results show that children in the dyslexic group differed significantly from the children in the control group consistently on the all eye-tracking measures, except on a few single colours for a few measures, namely fixation count (blue, turquoise O, purple O), fixation frequency (yellow), fixation duration average (yellow, orange, turquoise O, purple O), saccade count (blue, orange, turquoise O, orange O, purple O), saccade frequency (yellow), saccade duration total (blue, turquoise O), and saccade duration average (turquoise, red O, turquoise O). We observed significant differences between groups on EDA measure in all colours except yellow, turquoise O, and orange O. Children from the control group scored higher across the EDA measure. Regarding HRV measures, we observed significant differences between groups in mean RR (purple, turquoise O, yellow O) where dyslexic children scored higher and in mean HR (purple, turquoise O, yellow O) where the control group scored higher. Children with dyslexia showed significant increase of the beta band-power on purple and theta band-power on purple O colour in comparison to the control group, while the control group showed significant increase of the theta band-power on turquoise O colour.

Further on, for abetter overview, we showed a visual comparison between the two groups on every background/overlay colour for EEG (alpha, beta, delta, theta, broadband) in Figure 2; reading duration and eye-tracking measurements (fixation count, fixation frequency, saccade count, saccade frequency) in Figure 3; EDA in Figure 4; and HRV (mean RR and mean HR) in Figure 5.

More detailed analysis regarding comparisons of all the background/overlay colours with white across all of the parameters for both groups separately can be found in the Supplement.

Reading performance and physiological parameters comparisons (non-dyslexic vs. dyslexic children) over the average scores for all colours together are presented in Table 3. A significant difference was obtained regarding all eye/tracking measures, reading duration, and median beta power band, as well as for EDA, mean RR, mean HR, RMSSD, and CVRR. We observed no significant difference between non-dyslexic and dyslexic children in all other measures.

Dyslexic children demonstrated longer time duration for the reading performance in comparison to non-dyslexic children (control group). They also showed a higher fixation count, fixation duration total, fixation duration average, saccade count, saccade duration total, and saccade duration average, while the control group showed higher scores of fixation frequency and saccade frequency. Children with dyslexia also showed increased in all of the EEG bands except theta band in comparison to the control group. EDA was higher in the control group as well as mean HR, while all other HRV parameters were higher in dyslexic children.

## 4. Discussion

### 4.1. Summary

In the 36 children (18 dyslexic and 18 control group), we evaluated the relationship between physiological parameters and colour modifications in the text, background, and overlays during reading performance regarding reading duration, eye-tracking, EEG bands, EDA, and HRV parameters using simultaneously monitored sensor signals. The findings of the present study regarding reading duration on different background and overlay colours, as well as on white background with black text, show that there is a significant difference between groups. Firstly, dyslexic children took longer to read the text. Concerning the eye-tracking measures, we found significant differences between groups in all measures except fixation duration average for reading on white background with black text. Dyslexic children showed higher values for fixation count, fixation duration total, fixation duration average, fixation duration average, saccade count, saccade duration total, and saccade duration average, while non-dyslexic children scored higher on fixation frequency and saccade frequency, which is in agreement with the previous studies [56,57,58,59,60] where it was reported that dyslexic children have longer fixation duration, reading time, and saccade duration. It was also found that the control group scored higher on fixation frequency and saccade frequency in reading on white as well on coloured background/overlay, which is aligned with previous research where it was reported that typical readers simultaneously process the number of letters and adapt to them according to the task, while dyslexic children only process a few letters during the reading task [61,62,63]. Furthermore, older or better readers need more time to adapt to an unexpected text [64,65].

### 4.2. EEG Parameters

In the literature regarding EEG power bands (alpha, beta, delta, theta, and broadband activity), it is reported that dyslexic children have an increasing trend or disrupted oscillations in the range of delta and theta, and specifically increasing beta [66,67,68,69,70]. Regarding the broad band activity, it was reported that this measure could be used for distinction between dyslexic children and individuals with normal reading abilities [70]. In the present study, we found no significant differences in EEG bands between groups during the reading task on white background with black text, where dyslexic children showed an increasing trend in all EEG measures in comparison to the control group. Considering the reading performance in all colours together, we found significant increase values in the beta band-power in dyslexic children in comparison to the control group.

The present study gives further support to the reports in which it was found that colours have an important role in the reading process [23,31,32,33,34,35,36,37,41,44,46,47,48,71,72,73].

Regarding EEG, it was shown that during the reading on purple background colour, children with dyslexia had a significantly increasing trend of the beta and broadband activity and significantly increasing trend in the theta band while reading on purple overlay, which is aligned with previous studies [60,67,69]. The control group of children scored with increasing values of theta band on turquoise overlay colour in comparison to the group of dyslexic readers.

### 4.3. Eye-Tracking Parameters

The results of the present study regarding eye-tracking measures and reading in colours affirmed the previous study [44] where it was reported that the third colour from the selected colour set with the shortest reading duration in individuals with dyslexia was yellow, as is confirmed in the present study; however, our study shows that the first colour with shortest reading duration is turquoise and the second turquoise O, which is opposite to the previous study, where the first was peach and the second orange. Our study shows that the red colour has the longest reading duration in dyslexic children, followed by red O and yellow O. Regarding the control group, they scored with the shortest reading duration with violet O, blue O, and orange O, while the longer reading duration was recorded on blue, red O, and yellow O. The overall results regarding eye-tracking measures and reading in colour are aligned with the studies [35,74] where it was reported that influence of colours and coloured overlays is minimal for the readers with dyslexia regarding reading duration, except for the yellow and turquoise and turquoise O, as was previously reported [50,51]. Furthermore, the study confirms that there are systematic differences between groups on the white, single colours and all colours together, whereas the control group showed significantly higher values on the fixation frequency and saccade frequency, as was previously explained [61,62,63], while dyslexic children scored higher in all eye-tracking measures except for a few, namely fixation count (turquoise O, purple O), fixation duration average (orange, purple O), saccade count (blue, orange, turquoise, purple O), saccade duration total (blue, turquoise O), and saccade duration average (turquoise, red O, turquoise O).

### 4.4. EDA

Regarding electrodermal activity, it is important to mention that EDA is linearly correlated to arousal and reflects emotional response and cognitive activity [75,76], and it is one of the most-used psychophysiological measures for definition of stress level [77]. High electrodermal activity reflects a high level of stress. The present study reports that in both situations, reading on white and coloured overlays/background children with dyslexia scored significantly lower in comparison to the group of non-dyslexic children, as was also reported in previous studies [78,79]. TEDA measure could be used to distinguish children with and without dyslexia, and we can conclude that the control group had a higher stress level or emotional response to the colours, which is aligned with the report where it was found that good readers need more time to adapt to a new form of text [64,65] which may produce an emotional response in such situations [78,79,80]. Regarding additional colours, we have found that the EDA measure was significant in all colours between groups, except for orange, turquoise O, and orange O, which could confirm the previous reports that pastel colours have a calming effect [81,82].

### 4.5. HRV Parameters

A relationship between HRV parameters and modification in the text background and overlay in children with and without dyslexia is reported in the present study. During the reading on the white background with black text, which is typical for school-age children, we found no significant differences between dyslexic and non-dyslexic children. Our result gives further support to the previous studies where researchers reported that there is no systematic difference between dyslexic and non-dyslexic subjects regarding the HRV analysis [78,83]. However, considering all colours together during the reading task and physiological responses, we found systematic differences between dyslexic and non-dyslexic children in a few HRV parameters, namely, mean RR, mean HR, RMSSD, and CVRR. Dyslexic children have higher values in all named measures except mean HR, where the control group showed a higher score, which could also be related to the level of stress and aligns with the EDA measure, where the control group also showed higher scores [81,84,85]. Mean HR was significantly higher in the control group than in the dyslexic group for purple, turquoise O, and yellow O, where dyslexic children showed significantly higher values of mean RR. Regarding emotional response in dyslexic individuals [79], the level of arousal in our study was expected and found to be lower in the group with dyslexia, which is reflected also by the mean HR and mean RR measures on the single colours [84] and confirms the hypothesis where those measures could be beneficial for distinction between dyslexic and non-dyslexic individuals.

### 4.6. Colours Impact on Overall Measures

The aim of this study was to better understand the relationship between physiological parameters and colour modifications in text background and overlay during reading in children with and without dyslexia. The group of 36 school-age children (18 with and 18 without dyslexia) read the text on white and 12 background and overlay colours. The study evaluated differences between groups in the reading duration, EEG, HRV, EDA, and eye-tracking measures. Based on the findings, we can conclude that there are systematic differences between groups regarding reading duration and eye-tracking measures, whereby dyslexic children scored higher in overall results except for fixation frequency and saccade frequency, which is in line with the previous studies. Regarding single colours, where dyslexic children showed the shortest time duration to finish the reading task, we have found that reading on turquoise and turquoise O background colour was the easiest for them. In the third place was the yellow colour, which was also reported before [45]. Further, we have observed significant differences between groups regarding EDA measures, where the control group has higher values in comparison to the dyslexic one, which confirms the finding of the previous studies as well. Looking into the relation between single colours and EDA, we have observed no significant differences in orange, turquoise O, and orange O colours. Regarding HRV parameters, we found no significant differences in most HRV measures, which aligns with the previous studies, except for the mean HR, where the control group scored higher and which is related to the EDA and stress level as well. Also, systematic differences of the HRV measures were found in single colours for mean RR and mean HR in purple and turquoise O, where the control group scored higher on mean HR and dyslexic group on mean RR. In the previous literature, it was reported that a broadband EEG could be used for the distinction between dyslexic and non-dyslexic individuals. We found significant differences in the two groups on the purple colour, where dyslexic children showed an increasing trend, as well in the beta band for the same colour. Differences in brain waves between dyslexic and non-dyslexic children were explained in previous literature also by increasing of the beta band and oscillations in delta and theta bands. We have found significant differences between groups for turquoise O colour for the theta band, where the control group achieved higher values.

When compared across world languages, the main similarity found for dyslexia was the reading speed deficit (particularly in transparent orthographies [86], sometimes accompanied with other marks, such as a greater deficit for non word reading in comparison to the word reading, as well as an extremely slow and serial phonological decoding mechanism) in German and English [87] and frequency, orthographic neighbourhood size, and word length in Spanish and English [88], which leaves us with no clear sign of the underlying mechanisms of dyslexia. This is why some researchers have urged the identification of a robust sensory marker of phonological difficulties which would allow “early identification of risk for developmental dyslexia and early targeted intervention” [89], as well as the need to understand the pathophysiological visual and auditory mechanisms that cause children’s phonological problems [52]. Our study gives further support to the findings regarding reading duration in dyslexic children. However, it goes a step further in uncovering the cognitive and emotional patterns in dyslexic children through the implementation of a sensory hub developed for this study.

## 5. Conclusions

The goal of the study was to understand colour influence in reading performance and its relation to neurophysiological responses. By combining different modalities, we attempted to find a more objective approach to distinguish children with dyslexia from those without, in order to facilitate prevention through early detection or finding beneficial colour setup which could improve their reading performance. Based on the present findings, we can conclude that turquoise background, turquoise overlay, and yellow background could be beneficial for children with dyslexia and that all presented measures (eye-tracking, EDA, HRV, EEG) could be beneficial for the purpose of better understanding of the neurophysiological origin of dyslexia in children.

We believe that this is only the beginning of this line of research. The next step would be to try to adjust the preferred colour for each individual child, given that we found a large individual variation in the data set presented in this manuscript. By doing so, we would hope to make reading easier for children with reading difficulties and for the dyslexic children. Also, we plan to run machine-learning algorithms in order to assess which of all parameters (combinations of parameters) that were measured in this study give the best prediction of a particular child belonging to the dyslexic or non-dyslexic group. This would be very valuable for both prevention and early detection of dyslexia.

## Figures and Tables

**Figure 1 brainsci-11-00539-f001:**
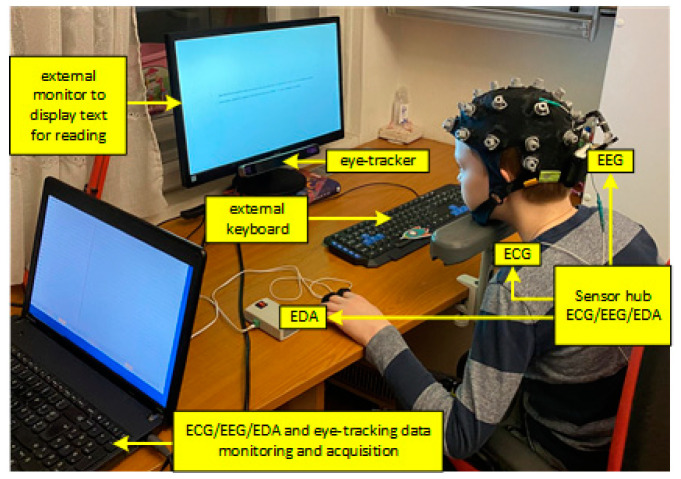
Sensor hub ECG/EEG/EDA computer for data acquisition, external keyboard and monitor, and eye-tracking system.

**Figure 2 brainsci-11-00539-f002:**
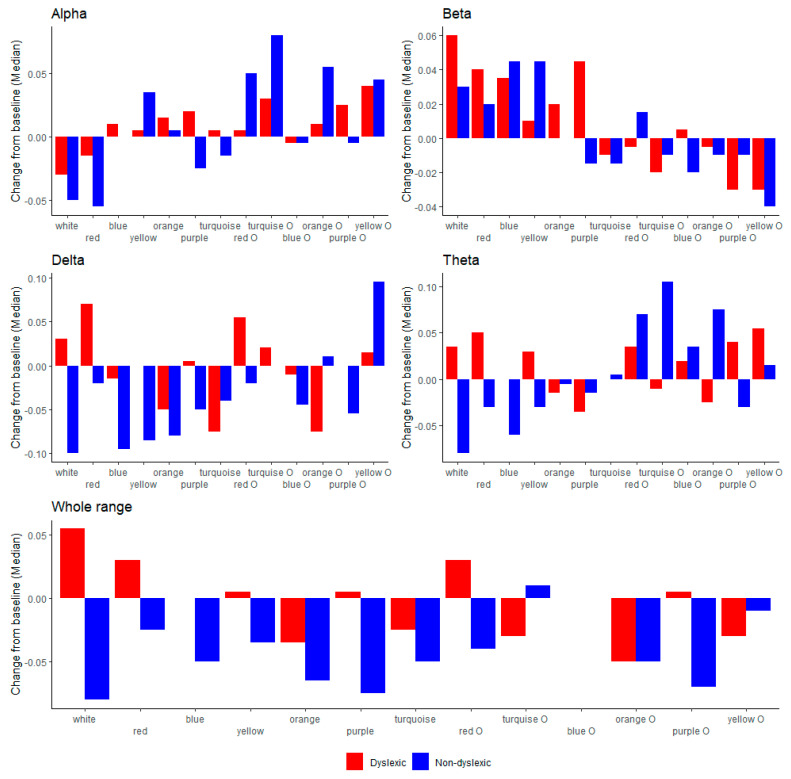
EEG (alpha, beta, delta, theta, broadband) between dyslexic and non-dyslexic children on every background/overlay colour.

**Figure 3 brainsci-11-00539-f003:**
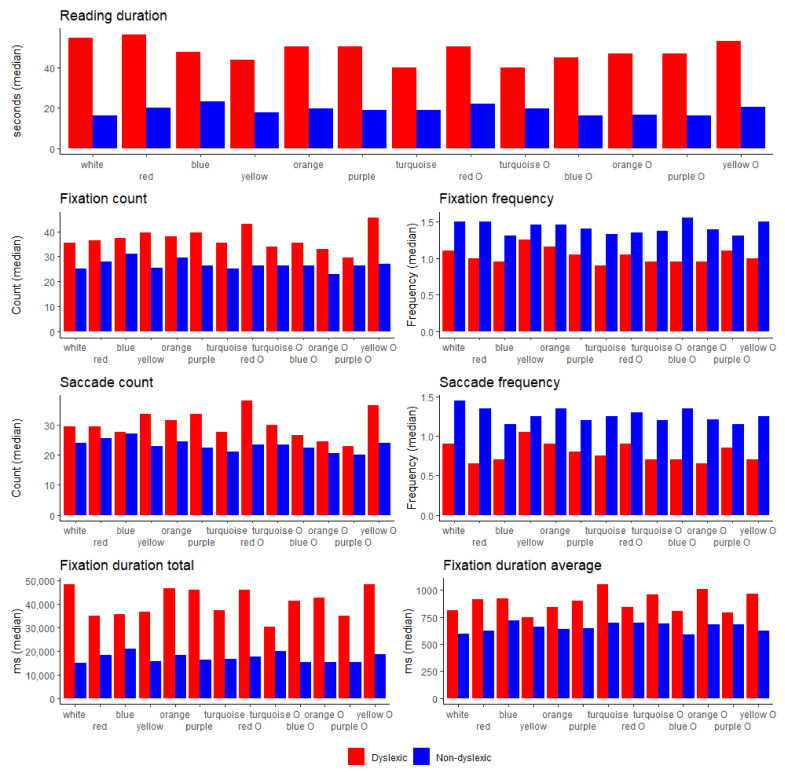
Reading duration and eye-tracking measures (fixation count, fixation frequency, saccade count, saccade frequency, fixation duration total, fixation duration average) between dyslexic and non-dyslexic children on every background/overlay colour.

**Figure 4 brainsci-11-00539-f004:**
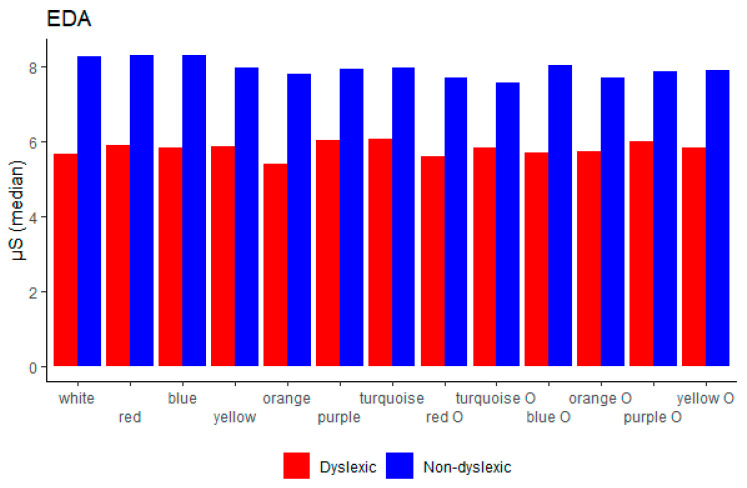
Electrodermal activity (EDA) between dyslexic and non-dyslexic children on every background/overlay colour.

**Figure 5 brainsci-11-00539-f005:**
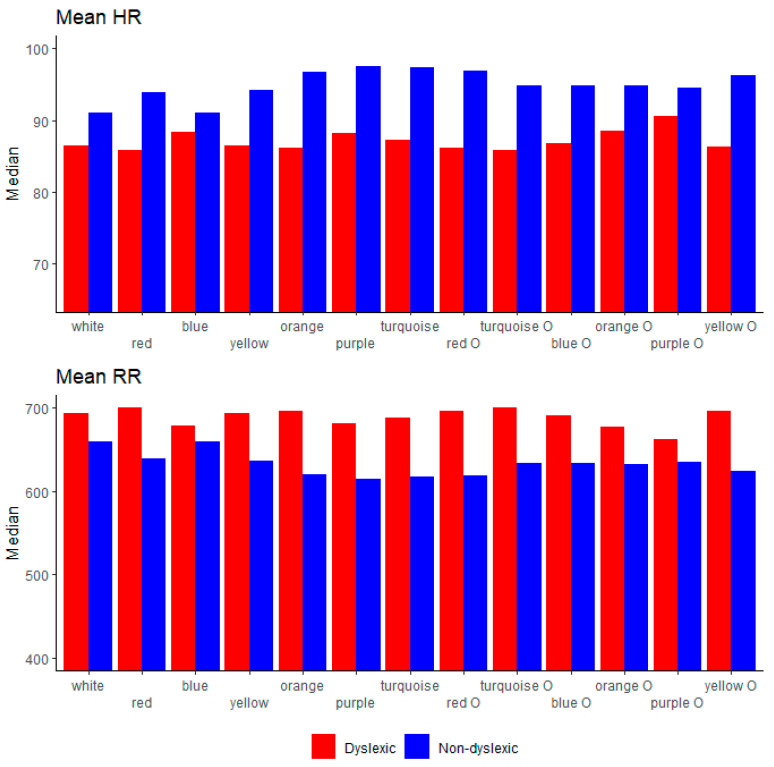
HRV parameters (mean HR, mean RR) between dyslexic and non-dyslexic children on every background/overlay colour.

**Table 1 brainsci-11-00539-t001:** Eye-tracking, EEG, EDA, and HRV parameters in non-dyslexic and dyslexic children; significant *p* values are marked as bold.

Parameters	Reading		*p* Value *
	Non-Dyslexic (*n* = 18)	Dyslexic (*n* = 18)	
Reading duration			
RD (s)	16.4 (11.9–23.5)	54.6 (26.1–70.7)	**0.002**
EEG parameters (relative band-power) ^a^			
Alpha	−5 (−11 to +9)	−3 (−6 to +8)	0.772
Beta	3 (−7 to +13)	6 (−2 to +13)	0.477
Delta	−10 (−17 to +30)	3 (−11 to+18)	0.809
Theta	−8 (−13 to +8)	3 (−7 to +9)	0.296
Broadband	−8 (−14 to +26)	5 (−13 to +17)	0.851
Eye-tracking parameters			
Fixation count	25 (23–30)	35.5 (29–67)	**<0.001**
Fixation frequency (count/s)	1.5 (1.37–1.80)	1.1 (0.6–1.6)	**0.036**
Fixation duration total (s)	15.1 (10.8–21.6)	48.3 (23.9–59.4)	**0.001**
Fixation duration average (ms)	593.8 (518.2–696.5)	809.7 (575.8–1586.9)	0.071
Saccade count	24 (20–28)	29.5 (25–42)	**0.003**
Saccade frequency (count/s)	1.45 (1.28–1.60)	0.90 (0.50–1.40)	**0.013**
Saccade duration total (ms)	469.7 (434.1–538.2)	736.5 (583.8–1509.7)	**<0.001**
Saccade duration average (ms)	20.1 (18.9–21.7)	23.1 (21.0–33.0)	**0.004**
EDA value			
EDA (μS)	8.29 (6.04–12.01)	5.67 (4.67–7.47)	**0.012**
HRV parameters			
Mean RR (ms)	659.4 (596.2–705.5)	693.6 (645.4–759.4)	0.092
STD RR (ms)	39.7 (22.2–57.3)	42.6 (32.5–59.9)	0.631
Mean HR (beats/min)	91.0 (84.4–100.6)	86.5 (79.0–92.9)	0.244
STD HR (beats/min)	6.28 (3.66–7.61)	5.22 (4.47–6.78)	0.988
RMSSD (ms)	50.9 (26.2–77.6)	44.7 (33.5–76.2)	0.527
CVRR = SDRR/MeanRR (ms)	0.08 (0.04–0.10)	0.07 (0.06–0.09)	0.828

Results are presented as median (25–75 percentile). * Mann–Whitney U test (exact *p* value). ^a^ EEG parameters are presented as difference from baseline (minus is decreasing trend from thebaseline while plus is increasing trend from the baseline).

**Table 2 brainsci-11-00539-t002:** Differences between dyslexic and non-dyslexic children on reading duration. EEG, eye-tracking, EDA, and HRV parameters, (*p* < 0.05 is marked with peach colour where dyslexic children scored with higher values and green colour where non-dyslexic children scored with higher values). The reported statistics is based on Mann–Whitney U test.

Colours	Red	Blue	Yellow	Orange	Purple	Turquoise	Red O	Turquoise O	Blue O	Orange O	Purple O	Yellow O
Reading duration (ms)	0.002	0.008	0.004	0.001	0.002	0.002	0.001	0.002	0.002	0.001	0.002	0.001
Fixation count	0.001		0.002	0.046	0.001	0.010	0.001		0.015	0.024		0.001
Fixation frequency (count/s)	0.008	0.005		0.041	0.018	0.003	0.026	0.011	0.011	0.046	0.016	0.010
Fixation duration total (ms)	0.002	0.016	0.004	0.001	0.002	0.002	0.001	0.031	0.003	0.002	0.005	0.001
Fixation duration average (ms)	0.058	0.043			0.043	0.005	0.050		0.058	0.050		0.025
Saccade count	0.018		0.005		0.001	0.057	0.002		0.049			0.001
Saccade frequency (count/s)	0.008	0.004		0.020	0.004	0.003	0.036	0.009	0.011	0.015	0.013	0.004
Saccade duration total (ms)	0.001		0.003	0.058	0.001	0.025	0.001		0.010	0.023	0.056	0.001
Saccade duration average (ms)	0.020	0.015	0.021	0.052	0.001				0.034	0.031	0.040	0.004
Alpha												
Beta					0.010							
Delta												
Theta								0.004			0.048	
Broadband					0.012							
GSR (uS)	0.029	0.053		0.038	0.022	0.057	0.041		0.049		0.053	0.035
Mean RR (ms)					0.047			0.040				0.027
STD RR (ms)												
Mean HR (beats/min)					0.047			0.040				0.027
STD HR (beats/min)												
RMSSD (ms)												
CVRR=SDRR/MeanRR (ms)												

**Table 3 brainsci-11-00539-t003:** Reading duration, HRV, eye-tracking, and EDA parameters in non-dyslexic and dyslexic children across all colours together (18 × 13 colours = 234 readings); significant *p* values are marked in bold. In Table 3, all the colours were averaged together and compared between the two groups on each measure.

Parameters	Reading		*p* Value *
	Non-Dyslexic (*n* = 18 × 13)	Dyslexic (*n* = 18 × 13)	
Reading duration			
RD (s)	21.63 ± 15.30	49.20 ± 29.81	**<0.001**
EEG parameters (relative band power)			
Alpha	2.5 ± 12.2	1.8 ± 11.4	0.529
Beta	−0.9 ± 16.5	5.1 ±27.5	**0.005**
Delta	5.9 ± 38.1	9.1± 47.5	0.415
Theta	2.7± 15.0	2.2± 16.4	0.735
Broadband	1.1 ± 29.1	6.2± 36.2	0.101
Eye-tracking parameters			
Fixation count	27.20 ± 08.50	45.29 ± 29.59	**<0.001**
Fixation frequency (count/s)	1.53 ± 0.54	1.15 ± 0.70	**<0.001**
Fixation duration total (s)	20.16 ± 14.57	43.57 ± 27.02	**<0.001**
Fixation duration average (ms)	697.20 ± 297.21	1108.41 ± 786.90	**<0.001**
Saccade count	24.33 ± 7.85	35.63 ± 23.65	**<0.001**
Saccade frequency (count/s)	1.37 ± 0.52	0.97 ± 0.69	**<0.001**
Saccade duration total (ms)	517.22 ± 189.82	1122.51 ± 1514.83	**<0.001**
Saccade duration average (ms)	21.34 ± 3.51	27.82 ± 12.11	**<0.001**
EDA value			
EDA (μS)	8.86± 3.77	6.30 ± 2.36	**<0.001**
	HRV parameters		
Mean RR (ms)	652.47± 96.37	690.61 ± 75.06	**<0.001**
STD RR (ms)	44.79 ± 37.31	47.49 ± 20.25	0.331
Mean HR (beats/min)	93.54 ± 11.94	87.89 ± 9.47	**<0.001**
STD HR (beats/min)	5.70 ± 3.07	5.96 ± 2.09	0.283
RMSSD (ms)	54.21 ± 48.26	56.71 ± 31.06	**0.001**
CVRR=SDRR/MeanRR (ms)	0.07 ± 0.04	0.08 ± 0.02	**<0.001**

* Independent sample *t* test. Results are presented as mean ± standard deviation.

## Data Availability

Data of this research is available upon request via corresponding author.

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
