# Peer review of "The Relation between Physiological Parameters and Colour Modifications in Text Background and Overlay during Reading in Children with and without Dyslexia"

_brainsci, 2021, doi:10.3390/brainsci11050539_

Round 1
Reviewer 1 Report
see attached document

Author Response
Thank you very much. Indeed, we believe that this is only the beginning of this line of research. The next step would be to try to adjust the preferred colour for each individual child, given that we found a large individual variation in the data set presented in this manuscript. By doing so, we would hope to make reading easier for children with reading difficulties and for dyslexic children. Also, we are planning to run machine learning algorithms in order to assess which of all parameters (combinations of parameters) that were measured in this study give the best prediction of a particular child belonging to a dyslexic or non-dyslexic group. This would be very valuable for both the prevention and early detection of dyslexia. We have now added this in the manuscript as you can find on the Page 15.
Reviewer 2 Report
Your article is very well written and shows scientific information regarding your trial yet it would be interesting to add further perspectives in your conclusions section, since the readers might be interested in knowing if you have the intention of expanding your experiment, i.e.: re-doing it with a greater amount of students, re-doing it at another university, etc.
Author Response
Dear Reviewer, thank you very much for your useful and thorough feedback and suggestions. We believe that, based on your suggestions, the current version of the manuscript has been improved substantially in comparison to the original version we submitted to the journal.
Point 1:
Several points need clarification/improvement before publication.
Introduction: you did not mention several studies testing effects of colored filters in dys (ex: Ray et al. 2005; Hall et al. 2013, Kim et al. 2015 (irlen syndrome), Razuk et al. 2018). Also the review de Stein 2018 is not cited.
Response 1:
We have included all the above suggested references in the introduction and in the discussion.
There are several more studies that took into account the effects of colour filter on reading in dyslexic children. For example, the reduction in Meares Irlen Syndrome symptoms was found [51], because of application of colour filters which eased the visual discomfort in these children. Furthermore, coloured filters were then considered as an effective intervention for delayed readers who had experienced visual stress. It was reported [52], that the reading speed of patients with Meares-Irlen syndrome improved by more than 20% when they wore the selected color-tinted lenses. By contrasting no filter, yellow and green filter, researchers [53,54] found that children with dyslexia were fastest and had the shortest fixation time, when reading in the yellow and green-filter condition.
Recently, Stein [55] came up with the argument that dyslexia is "characterized by poor temporal processing, hence impaired visual and auditory sequencing, that is caused by impaired development of transient/magnocellular (M-) systems throughout the brain" and thus needs to collect evidence not only from psychophysical tests, but from electrophysiological, eye movement, attentional, imaging, interventional and genetic findings as well.
You will find them on the Page 3 in the introduction and on the Pages 13,14 in the Discussion section.
Point 2:
Methods:
Subjects: a table showing clinical characteristics of 2 groups of subjects will help the reader. Inclusion and exclusion criteria need to be improved; what do you mean for…diagnosed with dyslexia, and that they have a normal, or corrected to normal vision?
Response 2:
The children who were in the experimental group are all officially diagnosed as dyslexic by the specialist in the field and were included in the study based on that criterion. On the other hand, kids in the control group were matched according to the age and gender, from the same city, but with no reading difficulties as assessed by their teachers. We made sure that this is more clearly stated as it can be seen on the Page 4 of the manuscript. So, the reading duration was not treated as one of the inclusion characteristics, but rather as a dependent variable, which was indeed higher in the dyslexic children in comparison to non-dyslexic children regardless of the background colour on which we presented the text. Also, normal, or corrected to normal vision meant that they either had no eye-sight problem, or if they did, they had adequate glasses, so that they could read the text with no difficulty. In fact, only one such child (with corrected vision) took part in the study. All the other children had no eye-sight difficulties.
Point 3:
How did you test attention disorders?
Response 3:
Many kids with dyslexia have accompanying disorders including attention problems. But, rather than testing these potential problems (such as attention) with behavioural tests, we believed that their cognitive specificities and differences will be captured by the fine-grained apparatus we employed in the study. For the attention, in particular, as it is obvious from the results it was apparent through the eye-tracking measures. Dyslexic children had systematic differences in comparison to the neurotypical children, and the pattern of the results differed based on the background colour.
Point 4:
Figure 1 shows the experimental set up; however you need to specify the type of eye tracker used its precision etc…. how you calibrated eye movements; how many electrodes the EEG system used.
Response 4:
SMI RED-m 120-Hz portable remote eye tracker (https://www.smivision.com). The SMI software was used for stimuli presentation (Experiment Centre 3.7), data collec-
tion (iView RED-m), and data analysis and visualisation (BeGaze 3.7)
We have now corrected the details that were missing from the description of the experimental set up as you will find on Page 5.
Point 5:
Reading task:
How did you select the reading text? It was matched with respect to reading age of children? I hope text was different for each colored filter condition but similar in term of words frequency, length of the word, number of words, etc…. You need to precise the text used and its psycholinguistic characteristic given that it is well known that words characteristic affect oculomotor pattern during reading.
Response 5:
The reading text was selected from the textbook for the third grade, so that it did not contain any of the words that kids would not be familiar with. Also, this text was selected because the paragraphs were comparable based on the length and complexity (per slide). Even so, we did pseudo-randomisation of the colours (so that the same O and T colour would not appear one after the other), but every paragraph would be seen by different subjects on the different background colour. This way, we made sure that the particular differences which you’ve rightly pointed out (such as word frequency, length of the word, number of words etc), would be averaged out.
We have made this more clear on the Page 4.
Point 6:
Statistical analysis:
Why you used the t-test instead of the ANOVA, given that with ANOVA you could compare the different colored filters used in the two groups of children tested?
Response 6:
We have corrected this in the current version of the manuscript and you will see the graphs for each of the measurements comparing the two groups across all the different measurements. Also, in the original manuscript we did not emphasize that for all measurements we had to use non-parametric statistics due to the data distribution, in the same way as it was reported in Table 1 for white colour. In Table 2 we compared dyslexic and non-dyslexic for each background/overlay colour so that the significant results which were higher in the dyslexics were marked with green colour and results which were higher in the non-dyslexic were marked with orange colour. Thus, because of the non-normal distribution of the data, all the analysis was run based on the Mann-Whitney U test. Again, we failed to state it explicitly in the original version of the manuscript, as we did for the first table.
Point 7:
Data processing:
I did not found the eye movements data. Which parameter did you analyzed? Results:
Response 7:
We have analysed the eye-movements data for all parameters, but in the table we reported only the significant ones. But, in the present version of the manuscript we reported all the measurements.
Point 8:
Table 3: legend need to be improved. Differences were small. You need to apply ANOVA (multiple comparisons) on the 2 groups of subjects testing the different parameters recorded in the different colored filters condition. Actually this table is very hard to understand. Could you make distinct graphs on oculomotor, EEG and EDA and HRV data?
Response 8:
We improved Table 3 and the legend. Also, we made distinct graphs for the EEG, eye-tracking, EDA and HRV measurements contrasting dyslexic and non-dyslexic children separately for each background/dyslexic colour after the Table 2. In Table 3 all the colours were averaged together and compared between the two groups on each measurement. This way, and based on the data distribution, it was possible to run parametric tests. Hopefully, you will find this way of presenting results more satisfying. We believe that many results have become more obvious when presented this way.
Point 9:
Reading duration: how did you measure it? This parameter is in the inclusion criteria so it should be moved in clinical table.
Response 9:
This parameter was not the inclusion criteria and we measured it for each child from the moment they’ve started reading till the moment they’ve finished the reading of the text, per each slide. As stated in the manuscript, the task was designed as self-paced, so kids would move from one slide to another in their own time.
Point 10:
Describe results on the basis of anova by comparing 2 groups in the different colored filters condition.
Response 10:
As previously stated in one of the comments above, we did the comparison between the two groups in the different colour filter conditions, which is shown in Table 2 and accompanying graphs (with non-parametric tests) and the we did the analysis you suggest here, too, and you will will the results of this analysis in the Supplement Table mentioned on the Page 11 (comparing each colour filter with withe for overall measures for the both groups separately) .
Point 11:
Discussion:
Please summarize your findings.
The data on reading time longer for dys than for non-dys is not a finding it is an inclusion criteria. So it should not be written here.
Re-organize the discussion in 4 sections following your results
Eye movements / EEG / EDA/HRV
Response 11:
We have summarised and organised the discussion in 4 sections marking them with the subtitles, as you will see on the Pages 11-13. However, we kept the reading duration in the discussion, as it was not the inclusion criterion, but the dependent variable in our study.
Point 12:
About your first finding (oculomotor data): it is well known that dys show an abnormal oculomotor pattern during reading and you miss several studies from several other countries (French, Italian, Greek, German…..).
Response 12:
We agree with this comment and we did include the suggested literature in the discussion, Page 13.
When compared across word languages, the main similarity found for dyslexia was the reading speed deficit (particularly in transparent orthographies [90], sometimes accompanied with other marks (such as a greater deficit for nonword reading in comparison to the word reading, as well as an extremely slow and serial phonological decoding mechanism" in German and English [92]; frequency, orthographic neighborhood size and word length, in Spanish and English [93], which leaves us with no clear sign of underlying mechanisms of dyslexia. This is why, some researchers urged the identification of a robust sensory marker of phonological difficulties which would allow "early identification of risk for developmental dyslexia and early targeted intervention" [94], as well as the need to understand the pathophysiological visual and auditory mechanisms that cause children's phonological problems [55]. Our study gives further support to the findings regarding reading duration in dyslexic children. However, it goes a step further in uncovering the cognitive and emotional patterns in dyslexic children through implementation of a sensory hub developed for this study.
Point 13:
The important goal of the study is to compare different colored filter and the effect of filters on oculomotor pattern. However you did not make the correct statistical analysis for that. So please run ANOVA and tell us if you find a different oculomotor behavior in DYS depending to the colored filter used; also such filtered did a change on non-dys group? I can not find this analysis.
Response 13:
We have run the suggested analysis as you will see in the Supplement Table and included it in the interpretation of the results in the discussion session. However, as stated above, some of it was based on means and some of it on medians, depending on the data distribution.
Point 14:
Second finding (EEG data):…. beta range in dyslexic children in comparison to the control group. This is interesting if ANOVA reports the same significant result. Add discussion on this finding, not just mention that other authors reported similar results. How do you explain this finding?
Response 14:
We did indeed find this result significant when we run ANOVA as well. The result was significant not only for purple, but for the red colour as well. We have previously run a non-parametric test, due to the distribution of the data, so we believe that this is why the results are not fully consistent (for the red colour there were more extreme values in both groups for instance). You will find this result in the additional document we’re sending you, so that you can compare the two results. Of course, we kept the non-parametric tests in the manuscript itself.
Point 15:
The conclusion section is too long, you can move some text in the Discussion section.
Response 15
We moved some text from the Conclusion to the Discussion section as you will see on the Page 13.
Point 16:
Finally, I can not understand your hypothesis driven of this study, and I think you have to write it clearly at the end of the Introduction: why do you expect to find difference in all these 4 measures (Eye movements / EEG / EDA/HRV) by using different colored filters? Add more thinking about that… I do not think that you can write that such measures can be clinically used for screening Dys vs non-Dys…An easy task for discriminate dsy vs non-dys is just reading text and count the time and the number of errors… or at least to measure eye movements because you can find easy eye trackers (remote system etc..) that can do that. Try to find another study to tell us, eventually just present data dys vs non-dys while reading a text-only, without taking into account the different types of colored filters.
Response 16:
Thank you, you have made us think more and rephrase this section. In fact, we expected that kids who are good readers, won’t mind the change of the colour, when measured through the reading duration. But, we expected that there will be more variation in that respect with dyslexic children. However, in both groups, we expected that cognitive and emotional arousal will vary with the change of the colour and that could only be measured by employing a fine-grained measurement which is not prone to the strategy that kids may use but are rather automatic measures of their cognitive and emotional engagement in the task. We expected that the pastel colours would focus their attention better and make the reading task easier for them, unlike strong colours, which we expected would be challenging to read on, to both dyslexic and non-dyslexic children. These parameters make it possible to conclude which colours would result with the better focus on the text, which otherwise could not be measured with rough measures such as reading duration. We have made our expectations clear at the end of the introduction section as you can see on Page 3.

Round 2
Reviewer 1 Report
authors did not take in account my previous comments

Author Response
Dear Reviewer, we believe that, based on your suggestions, the current version of the manuscript has been improved substantially in comparison to the previous version.
Point 1:
The authors have to respond points by points to questions/changes asked by the reviewer
End of the Introduction: Explained why you wrote this sentence:…. We expected that the pastel colors would focus their attention better and make the reading task easier for them, unlike strong colors, which we expected would be challenging to read on, to both dyslexic and non-dyslexic children.
Response 1:
We wrote that sentence as a response to your request in the previous round: “I think you have to write it clearly at the end of the Introduction: why do you expect to find difference in all these 4 measures (Eye movements / EEG / EDA/HRV) by using different colored filters?”).
We have further elaborated this in the following paragraph of the introduction :
Reading involves sensory integration, attention and memory processes which are reflected in the psycho-physiological states of the individual and measurable by different biosignals. Our previous work introduced a sensor hub for measurement of four biosignal modalities, and describes the contributions of each modality to understanding of different aspects of neural, physiological, and behavioral processes in children during reading [22]. Based on the previous reports on dyslexia [55,56] and the results of our work on reading in children [22], the main purpose of the present study is to apply previously validated set of sensors and extend established experimental protocol to children with dyslexia in order to better understand the neurological basis of this disorder and its relation to the colours.
We have also modified the sentence describing our hypothesis and backed it up with references :
We expected that the pastel colors would facilitate the reading task, unlike intense colors, which we expected would be more challenging, both to dyslexic and non-dyslexic children, based on previous reports [36,45].
Point 2:
The point on subjects selection from my first revision was not completely improved.
Please do it as previously suggested give information on selection of subjects. The rage is also missed both in abstracts as well as methods section
Response 2:
The main contribution of our paper was to investigate the impact of colors on physiological parameters acquired during the reading of children with and without dyslexia. For both groups (with and without dyslexia) we consulted the experts that work with both groups for a long time period. For the children without dyslexia, we consulted school teachers and school speech therapists, and then we have selected children for the control group matching the age and gender with a group of dyslexic children. The children with dyslexia were selected from the group which is already in the treatment process at the speech therapist, who has diagnosed dyslexia to each child individually according to the standard criteria that are used in daily practice (speed of reading, accuracy, understanding, and human intelligence). Only if the discrepancy between the three above-mentioned tests and IQ measurements were larger than the 1.5 standard deviation, the children were included in the study. In fact, although the criterion was set to a minimum of 1.5 SD, in each case it was more than 2 sd.
We have now added this clarification to the methods section, subsection - subjects: The dyslexic children in this study have already undergone adequate speech treatment for dyslexia. All of them satisfied the standard criteria for dyslexia diagnosed by the speech therapist based on three different tasks: speed of reading, accuracy (phonological decoding) and understanding measured on the standardized text adapted for this age in Serbian. Also, the IQ of children was tested on the Raven's Progressive Matrices and only those who scored higher than 90 IQ were selected for the study.
We are sorry we have missed adding the age of the children. We have now added it in the subjects section and in the abstract of the paper.
Point 3:
Make a data processing for oculomotor, EEG and EDA and HRV data; actually what did you analyzed for oculomotor data is missing
Response 3:
At the beginning of the section Experiment design, we emphasised that “Experimental design was exactly the same as in the [22] except for the difference laid out in the description of Figure 1 and Data processing part.” Hence, the eye-tracking data has not been changed and because of that we did not mention them.
The full extent of the description of the eye-tracking data was therefore covered in our previous research (22), as you can see in the description that follows. All of the below listed parameters you can find in Table 1, Table 2 and Table 3 in the present research.
Eye tracking data analysis and visualization was performed using SMI BeGaze TM 3.7 software (SensoMotoric Instruments, Teltow, Germany). The selected eye-tracking parameters were fixation count, fixation frequency (count/second), fixation duration total (ms), fixation duration average (ms), saccade count, saccade frequency (count/second), saccade duration total (ms), and saccade duration average (ms).
If you considering that is better to reorganize this section, ant to present all the data in the present Article as well, please emphasize.
Point 4:
Again the authors did not correct the question concerning statistical analysis used. Please do it and respond to my question. Why you did not do an ANOVA.
Response 4:
We were attending to use ANOVA as an appropriate method for multiple colors and two groups. But the problem with the data is high variability, small sample size (exploratory study) and absence of normality of distribution. Since ANOVA is very flexible, we were, even, exploring the data with histograms and QQ plots (not using any tests for normality of data), but the small sample size and high variability suggests that nonparametric alternatives should be used as a more appropriate. In nonparametric methods, extreme values have no influence as they do in parametric methods. Additionally, the EEG parameters were measured using electrodes where the signal might be with the noise (extreme values), and for that reason median values of signals were used as a central tendency measurement for each subject in each measurement. The nonparametric approach tests the difference between medians instead of means, and this is the reason why we believe that the nonparametric approach is more appropriate in this situation.
We think that using uniform approach might be better and, therefore, we used nonparametric tests though all exploratory parameters in the study when comparing nondyslectic and dyslectic 2x18 subjects. This includes HR and RR whose distributions are approximately normal.
As you suggested, we parallely did the ANOVA for repeated measurements with dyslexia as a fixed factor and colours as repeated measurements, but did not obtain any significance in overall testing. In post hoc testing we obtained significant differences regarding the groups and regarding the other colors vs. white. But, as we already mentioned, a nonparametric approach was considered as more appropriate, and since we are not using any advanced statistical methods, we decided to be as clear and simple as possible using the medians and nonparametric tests.
When we used gathered data (13 colors x 18 subjects) we used parametric tests because the number of subjects in the analysis is appropriate and the distribution is considered to be approximately normal (Table 3).
Point 5:
Figures you added: the legend needs to be improved. You have to add standard deviation or standard errors!
Response 5:
In these graphs, we presented median values. As you probably know, we can only add CI for medians or interquartile range, as a presentation of the variability. We tried to do that and the problem is that the graphs become overcrowded. When we have several graphs in one figure it becomes impossible to understand and focus on results. As an example we present you one figure to see how difficult it is to understand the results when an additional element, for example 95% CI is added.
Medians with 95% CI
*We could make changes on Figures and Legend like is presented above, if you still think that it is necessary.
We have also improved legends in the Results section.
Point 6:
Again about result on reading duration the authors did not correct. Please read again my previous comments and try to correct the ms following my suggestions.
Response 6:
The variable that we have named as reading duration (RD) in our paper, was extracted from our sensory hub system, more precisely from SMI eye-tracking system and it is a dependent variable, as well as other measures.
Reading duration is not the same variable as the speech therapist estimated (we have explained it above) in the individual tests before the experiment for dyslexia diagnosis.
Point 7:
About figures you added i do not understand the utility to add graphics on both count and frequency of saccades and fixations (it is the same?). Also, it is quite logic that a saccade is followed by a fixation. SO i think that the data on fixation is enough. In contrast given that the read of words is done during the fixation, the most important parameter (that also could be influenced by colored flirters) is the duration if fixation. So maybe you can add this in the graphic instead of the number of saccades.
Response 7:
Actually, fixation and saccade data and graphs are not redundant. Minimal duration of fixation in the system we used was set to 100ms (typically ranges from 150 to 300 msec). The definition of saccades and fixations are well described in the article (Is the eye-movement field confused about fixations and saccades? A survey among 124 researchers): “A fixation may, for example, be considered as a cluster of gaze coordinates within a specified range in space and time, whereas a brief peak in the velocity signal of the gaze signal may correspond to a saccade. Note that the thresholds for what constitutes a computational fixation or saccade may be determined by the dimensions of the text that was presented.” So, any eye-stops that last less than minimally defined fixation are not counted as fixation. On the other hand, Rayner states: ‘successive fixations that are on adjacent characters count as a single fixation’. Based on all the above stated, hopefully it is clear why the number of saccades and fixations are not equal/redundant. Finally, there are a number of papers which use exactly the same eye-tracking measurements as we reported in this study and we referred to some of them in this paper [57-61,62,63,64].
Finally, we have added two more graphs showing the fixation duration effects (fixation duration average and fixation duration total).
